# Use of Deep-Amplicon Sequencing (DAS), Real-Time PCR and In Situ Hybridization to Detect *H. pylori* and Other Pathogenic Helicobacter Species in Feces from Children

**DOI:** 10.3390/diagnostics14121216

**Published:** 2024-06-08

**Authors:** Yolanda Moreno Trigos, Miguel Tortajada-Girbés, Raquel Simó-Jordá, Manuel Hernández Pérez, Irene Hortelano, Miguel García-Ferrús, María Antonia Ferrús Pérez

**Affiliations:** 1Research Institute of Water and Environmental Engineering (IIAMA), Universitat Politècnica de València, 46022 Valencia, Spain; ymoren@upvnet.upv.es (Y.M.T.); ihormar89@gmail.com (I.H.); 2Department of Pediatrics, Obstetrics and Gynecology, University of Valencia, 46010 Valencia, Spain; simo_raq@gva.es; 3Department of Pediatrics, La Fe Polytechnique and University Hospital, 46026 Valencia, Spain; 4Foundation for Promotion of Health and Biomedical Research in the Valencian Region (FISABIO), 46020 Valencia, Spain; 5Department of Pediatrics, University Hospital Doctor Peset, 46017 Valencia, Spain; 6Biotechnology Department, Universitat Politècnica de València, 46022 Valencia, Spain; mhernand@btc.upv.es (M.H.P.); mferrus@btc.upv.es (M.A.F.P.)

**Keywords:** *Helicobacter pylori*, non-*pylori* helicobacters, real-time PCR, in situ hybridization, deep-amplicon sequencing (DAS), feces, children

## Abstract

Background: Detecting *Helicobacter pylori* in fecal samples is easier and more comfortable than invasive techniques, especially in children. Thus, the objective of the present work was to detect *H. pylori* in feces from children by molecular methods as an alternative for diagnostic and epidemiological studies. Methods: Forty-five fecal samples were taken from pediatric patients who presented symptoms compatible with *H. pylori* infection. HpSA test, culture, real-time quantitative PCR (qPCR), fluorescence in situ hybridization (FISH), direct viable count associated with FISH (DVC-FISH), and Illumina-based deep-amplicon sequencing (DAS) were applied. Results: No *H. pylori* colonies were isolated from the samples. qPCR analysis detected *H. pylori* in the feces of 24.4% of the patients. In comparison, DVC-FISH analysis showed the presence of viable *H. pylori* cells in 53.3% of the samples, 37% of which carried 23S rRNA mutations that confer resistance to clarithromycin. After DAS, *H. pylori*-specific 16S rDNA sequences were detected in 26 samples. In addition, DNA from *H. hepaticus* was identified in 10 samples, and *H. pullorum* DNA was detected in one sample. Conclusion: The results of this study show the presence of *H. pylori*, *H. hepaticus*, and *H. pullorum* in children’s stools, demonstrating the coexistence of more than one *Helicobacter* species in the same patient. The DVC-FISH method showed the presence of viable, potentially infective *H. pylori* cells in a high percentage of the children’s stools. These results support the idea that fecal–oral transmission is probably a common route for *H. pylori* and suggest possible fecal–oral transmission of other pathogenic *Helicobacter* species.

## 1. Introduction

*Helicobacter pylori* is considered the most common infectious agent in humans. Its ecological niche is the gastric mucosa, which persists for long periods and produces inflammatory-type gastritis and gastric ulcers, although its presence is asymptomatic in many cases. The organism has also been reported to be involved in developing 80% of all gastric cancers worldwide [1].

The global prevalence rate of *H. pylori* in children is estimated to be within 30%, much higher in developing countries (almost 60%) than in developed regions (less than 10%) [2,3]. Nodular gastritis is the main endoscopic finding of *H. pylori* infection in children, but gastroduodenal erosions/ulcers are also seen, especially after ten years of age. Infection in children differs from that in adults in that children rarely develop complications, and clinical manifestations are nonspecific. Moreover, the number of microorganisms in the gastric mucosa is frequently lower than in adults. However, *H. pylori* remains a significant risk factor for duodenal ulcers and gastritis in children. Infection is also related to other extraintestinal illnesses, such as immune thrombocytopenic purpura (ITP) and unexplained iron deficiency anemia (IDA) [4].

Following internationally standardized protocols, several eradication alternatives are applied sequentially to treat the infection. For children, treatment options are more limited. For those with gastric ulcers, chronic ITP, or refractory IDA, testing for *H. pylori* is recommended, and treatment should be based on the results of antimicrobial susceptibility testing. If antibiotic sensitivity is unknown, the first-choice option is triple therapy for 14 days [5].

However, there have been numerous failures in all therapeutic regimens due to the emergence of resistant strains. The leading cause of treatment failure in children is clarithromycin resistance, with rates that vary from 11.9% to 50.9% in different countries [6]. Resistance of *H. pylori* to clarithromycin is associated with some point mutations in nucleotide positions A2146 and A2147 of the 23S rRNA gene that can be detected by molecular methods [7].

Despite numerous efforts, the exact transmission mode of *H. pylori* has yet to be determined, and epidemiological studies suggest that fecal–oral transmission could be the most common transmission route, especially in developing countries [8]. The presence of *H. pylori* in feces has been demonstrated by culture, but fecal–oral transmission continues to be questioned due to the low concentration of *H. pylori* found in feces compared to other fecal pathogens of public health importance. Furthermore, the culture of *H. pylori* from stools has yet to be achieved. Although specific antigens and DNA are common, it is unclear whether *H. pylori* can remain viable during its transit through the enteric tract [9].

Several studies have linked other *Helicobacter* species (non-*H. pylori*-Helicobacter, NHPH), such as *H. hepaticus*, *H. pullorum*, or *H. bilis*, to gastrointestinal diseases, such as enteritis, hepatitis, hepatocarcinoma, chronic liver disease, cancer pancreas, cholecystitis and inflammatory bowel disease [10,11]. Fecal–oral transmission has been proposed for some of these NHPH species [12], and *H. pullorum* has been isolated from the feces of patients with and without diarrhea. However, more information and research should be needed regarding the prevalence, pathogenesis, diagnosis, and epidemiology of NHPH species [10].

*Helicobacter pylori* can be detected in clinical samples by invasive and noninvasive methods. Invasive techniques such as endoscopy and subsequent gastric biopsies for culture, histology, rapid urease tests (RUTs), or PCR are commonly used. Histology is the gold standard [1] for detecting *H. pylori* infection. However, it is not generally applied to children, and its use is recommended only for antibiotic resistance testing or if there are clear clinical indications for endoscopy [5].

Culture is 100% specific, demonstrates the existence of an active infection, and allows for determining bacterial susceptibility to antibiotics. However, it is a tedious, time-consuming procedure and unnecessary for the routine diagnosis of most patients. Moreover, sensitivity varies significantly among laboratories, from only 50% to 70% of infected biopsies [13]. Likewise, the culture and isolation of other *Helicobacter* species is highly complicated since there are no adequate and specific methods. The most frequent noninvasive diagnostic methods for *H. pylori* infection are the urea breath test, the detection of specific antibodies in serum, and antigen detection assays from stool, saliva, and urine. However, these techniques also present some problems, such as nonoptimal specificity or difficulty in assessing the existence of an active infection [1].

*Helicobacter pylori* detection in fecal samples is an easier and more comfortable practice than invasive techniques, especially in children, as feces can be obtained even without active collaboration. Moreover, these techniques could facilitate epidemiological studies on the mode of transmission of the bacteria [14].

The culture of *H. pylori* from feces is challenging, as feces contain many microorganisms, bile salts, polysaccharides, and degradative enzymes, which can hinder bacterial growth or survival. Bacteria may also be present in their viable but not culturable (VBNC) form due to the stressful environmental conditions along the intestinal tract [9]. Due to these disadvantages, *H. pylori* antigen (HpSA) detection has been used to detect *H. pylori* in stool samples. The main limitation of this method is that false negatives can be obtained. Furthermore, a positive result does not imply the viability of the pathogen, and antibiotic resistance tests or genetic characterization cannot be performed [14].

To improve both the efficacy of detection in feces and our knowledge about *H. pylori* transmission, alternative molecular methodologies have been developed, such as polymerase chain reaction (PCR) or in situ hybridization with fluorescent probes (FISH). PCR has been proposed as the most effective molecular method for detecting *Helicobacter* in clinical samples [15]. However, PCR cannot differentiate between living and dead cells since it can amplify genetic material from inactive cells and extracellular DNA [14].

The FISH technique detects nucleic acid sequences using a fluorochrome-labeled probe that specifically hybridizes with the complementary sequence of bacterial rRNA without damaging cell morphology [16]. Using the FISH technique, viable and nonviable cells are detected. Combining the direct viable count method with fluorescence in situ hybridization (DVC-FISH) can avoid this limitation, allowing for specific detection of viable cells. DVC-FISH has been previously used to detect and identify viable *H. pylori* cells in complex samples such as feces [17].

All the molecular analysis techniques described thus far need to provide information on the abundance or diversity of the significant microorganisms in the microbial community. Metagenomics is the only technique that allows these features to be studied through massive high-throughput sequencing (next-generation sequencing, NGS). This technique has been used to study the diversity of different aquatic environments [18].

Sequencing based on marker genes (deep-amplicon sequencing, DAS) is the most widely used technique to determine microbial diversity in complex environmental samples. In this method, DNA is initially extracted from the sample and amplified with a taxonomically informative gene common to all organisms of interest. This method is characterized by its analytical speed, great computational ease due to the lower number of sequences obtained, excellent efficiency in the reaction, and high taxonomic resolution. However, its main handicaps are the need to amplify the region to be studied by PCR previously, the fact that amplicon sequencing can produce very different diversity values depending on the primers used, and that chimeras (artificial sequences) can be produced because of sequencing errors [19]. One of the fundamental pillars of metagenomics is DNA sequencing. Currently, one of the most widely used massive sequencing platforms for the study of microbial communities is Illumina^®^. It generates more than 90% of the world’s sequencing data and is characterized by high accuracy; high error-free read performance, and a high percentage of base calls [19].

The leading utility of deep-amplicon sequencing for bacterial screening lies in its ability to provide detailed insights into the microbial composition and dynamics within a sample. By generating large amounts of sequence data, deep-amplicon sequencing enables the detection and characterization of non-culturable, fastidious, or slow-growing microorganisms that other screening methods in complex microbial communities [20] may miss.

Regarding *Helicobacter pylori*, targeted sequencing methodology has been used to investigate changes in the microbial community composition after *H. pylori* infection or to detect drug resistance mutations in *H. pylori* and their correlation with phenotypic resistance. However, to our knowledge, studies have yet to be carried out for diagnostic purposes [15].

Using sensible and specific molecular methods for detecting *H. pylori* in feces could be an alternative to invasive diagnostic methods, particularly for children. Moreover, it can provide information about the possibility of fecal–oral transmission. Thus, the objective of the present work was to determine the presence of *H. pylori* in the feces of children presenting gastric symptoms. Culture and molecular methods such as real-time PCR (qPCR), FISH, DVC-FISH, and Illumina-based NGS were directed explicitly to *H. pylori* to reach this goal.

## 2. Materials and Methods

### 2.1. Sampling

A prospective longitudinal study was conducted in a Caucasian Spanish population of 45 consecutively recruited children and adolescents between the ages of 2–15 years (male 22, female 23). They were referred by their primary care pediatrician to the outpatient Pediatric Gastroenterology Unit of University Hospital (Dr. Peset, Valencia, Spain) for health checks and family studies of abdominal pain, among other conditions, between June 2016 and June 2017. None of the children had received previous antibiotic treatment that could prevent the growth of the bacteria.

The children resided on the Spanish Mediterranean coast at a latitude of 39°28′48″. Written consent for inclusion in the study was obtained from parents and guardians and children > 12 years old. The study was conducted following the Declaration of Helsinki (revised in 2013), and the ethical committee approved it for the hospital’s clinical research (approval number CEIC 68/16).

The gastroenterology service of the hospital took forty-five fecal samples (one sample per patient), and eight patients were diagnosed with *H. pylori* infection by the stool antigen HpSA test (Table 1).

### 2.2. Processing of Fecal Samples

Samples were collected in sterile tubes, transported to our laboratory under refrigeration within 24 h, and treated without further delay. The entire detection protocol for *H. pylori* has been summarized in Figure 1.

Two grams of each sample of fresh feces were mixed with 20 mL of PBS and vortex-homogenized. Aliquots were taken for culture, FISH, DVC-FISH, qPCR, and NGS tests, as described below. All these samples were codified as direct samples (D). An additional aliquot of 10 mL from each homogenized sample was incubated in 20 mL of Brucella Broth enrichment media (BBLTM (Becton Dickinson, Frankin Lakes, NJ, USA), 5% (*v*/*v*) fetal bovine serum (PAA Laboratories GmbH, Colbe, Germany), and 3.5 µg/mL polymyxin B, and Dent selective supplement (Oxoid, Hampshire, UK) under microaerophilic conditions (5% oxygen, 10% carbon dioxide, and 85% nitrogen) in anaerobic jars (Campy Gas Pak system; Oxoid, Hampshire, UK) at 37 °C for 24 h. After incubation, aliquots were taken for culture, qPCR, FISH, DVC-FISH, and NGS analysis (enriched, “E” samples). All analyses were performed in duplicate.

### 2.3. Helicobacter Pylori Reference Strains and Culture Conditions

The *H. pylori* NCTC 11637 strain was used as a positive control. Bacteria were grown on Dent selective agar (Columbia agar plates; BD Difco Laboratories, Frankin Lakes, NJ, USA), 10% (*v*/*v*) horse defibrinated blood, 0.025% (*v*/*v*) sodium pyruvate, and Dent selective supplement (Oxoid, Hampshire, UK) at 37 °C under microaerobic conditions.

### 2.4. Culture

One hundred microliter aliquots of each direct and enriched sample were placed directly onto selective Dent agar, and 400 µL aliquots were placed onto a 0.65 µm cellulose acetate membrane filter (Whatman, Maidstone, UK) deposited on Dent agar. Plates were incubated under microaerophilic conditions at 37 °C for 24 h, and the membranes were removed. After that, the plates were incubated under microaerobic conditions and examined for the presence of *H. pylori* at 48 and 72 h and 5, 10, and 14 days. Presumptive colonies were examined by Gram staining and subsequently analyzed by specific *H. pylori* vacA qPCR, as described below.

### 2.5. Real-Time q-PCR

DNA was extracted from a 1 mL aliquot of each sample using the GeneJetTM Genomic DNA Purification kit (Fermentas, Vilnius, Lithuania) according to the manufacturer’s instructions, with an initial incubation of 30 min at 56 °C.

Two microliters of the extracted DNA were used as a template for SYBR Green-based real-time qPCR to amplify a 372-bp-specific fragment from the *H. pylori* vacA gene [21]. The reaction was performed in a final volume of 20 µL containing 2 µL of LightCycler Fast-Star DNA Master SYBR Green I (Roche Applied Science, Penzberg, Germany), 1.6 µL of MgCl_2_ (4 mM), 0.5 µL of each primer, and 2 µL of DNA template. The amplification consisted of an initial DNA denaturation step at 95 °C for 10 min, followed by a 40-cycle reaction of 95 °C for 10 s, 62 °C for 5 s, and 72 °C for 16 s, followed by one extension cycle at 72 °C for 15 s (LightCycler 2.0 Thermocycler, Roche Applied Science, Penzberg, Germany). DNA from *H. pylori* strain NCTC 11637 was used as a positive control, while a qPCR mix without DNA was used as a negative control. Samples in which all the tests (culture, PCR, and FISH) were negative were considered PCR-specificity controls during the study. Samples were considered presumptively positive for *H. pylori* if the cycle threshold (Ct) was less than or equal to 40 and showed Tm values between 84.8 and 86.5 (Figure 2). The detection was confirmed by 1.5% agarose gel electrophoresis stained with 0.01% GelRed (Biotium, Fremont, CA, USA) of the amplification products. In addition, amplicons from all the positive samples were purified with the GenElute PCR Clean-Up Kit (Sigma Aldrich, St Louis, Mo, USA) and subsequently analyzed by sequencing (IBMCP, CSIC; Valencia, Spain). The homology of the amplified sequences with the corresponding *H. pylori* vacA gene fragment was determined by BLAST alignment (www.ncbi.nlm.nih.gov/BLAST. Last accessed 2 February 2024).

### 2.6. In Situ Hybridization (FISH and DVC-FISH) Analysis

FISH and DVC-FISH analyses were performed according to Moreno et al. [22] and Piqueres et al. [23], respectively. For FISH, 1 mL of each sample was centrifuged at 8000 rpm at 4 °C for 8 min. The supernatant was removed, and the pellet was resuspended in 1 mL of 1X PBS, washed again, and fixed in a solution of 1:3 PBS/4% paraformaldehyde (*v*/*v*) for 1.5 h at 4 °C. Afterward, samples were centrifuged at 8000 rpm at 4 °C for 8 min and washed with 1X PBS three times. Finally, the pellet was resuspended in 1:1 PBS/ethanol (*v*/*v*). Samples were stored at −20 °C until hybridization.

For DVC-FISH analysis, another 1 mL of each sample was added to 9 mL of DVC broth (BBLTM Brucella broth (BAS, Becton Dickinson, Franklin Lakes, NJ, USA) supplemented with 5% fetal bovine serum (Thermo Fisher, Whalthman, MA, USA) and 0.5 mg/L of novobiocin) and incubated for 24 h at 37 °C under specific microaerophilic conditions. After incubation, the sample was centrifuged at 8000 rpm at 4 °C for 8 min and fixed with paraformaldehyde as described above for FISH. Samples were stored at −20 °C until hybridization.

For both FISH and DVC-FISH assays, fixed samples were hybridized with an LNA *H. pylori* 16S rRNA probe (HPY: 5′-CCTG GAG AGA CTA AGC CCT CC-3′). As a positive control, we used a mix of three EUB338 probes (EUB338-I: 5′-GCT GCC TCC CGT AGG AGT-3′; EUB338-II: 5′-GCA GCC ACC CGT AGG TGT-3′; EUB338-III: 5′-GCT GCC ACC CGT AGG TGT-3′), complementary to a region of the 16S rRNA of the Eubacteria domain [22]. As a negative control, in all the assays, we used a feces sample that had yielded previous negative results for *H. pylori* culture, *H. pylori* VacA and *Helicobacter* 16SrRNA PCR, and specific *H. pylori* and *Helicobacter* sp. probe FISH assays.

Another FISH assay detected *H. pylori* cells harboring 23S rRNA mutations that confer resistance to clarithromycin. The combination of probes used in this case was a CLAR probe mixture (CLAR-I: 5′-CGG GGT CTT CCC GTC TT-3′; CLAR-II: 5′-CGG GGT CTC TCC GTC TT-3′; CLAR-III: 5′-CGG GGT CTT GCC GTC TT-3′), according to Trebesius et al. [24]. All probes were synthesized and labeled with CY3 (HPY and CLAR) and fluorescein (EUB 338 16S rRNA) by EXIQON (Woburn, MA, USA).

For hybridization, 5 µL of each fixed sample was deposited onto one well of a diagnostic slide pretreated with 0.1% gelatin. Slides with fixed samples were dehydrated by serial immersions in 50%, 80%, and 100% ethanol for 3 min each. The wells were then covered with 10 µL of hybridization buffer (0.9 mL/L NaCl, 0.01% SDS, 20 mm/L Tris-HCl [pH 7.6]) containing formamide at a final concentration of 40% for the HPY and 30% for the CLAR probes, together with 50 ng of each probe, and maintained at 46 °C for 2 h. Afterward, the slides were incubated with 50 mL of washing solution (0.10 M NaCl, 0.02 M HCl-Tris, 0.01% SDS, and 0.005 M EDTA) at 48 °C for 15 min. Finally, the slides were washed with distilled water and air-dried. All hybridization processes were performed under dark conditions. For all analyses, each slide used a fixed pure culture of the *H. pylori* NCTC 11637 strain as a positive control.

Slides were mounted with FluoroGuard Antifade Reagent (Bio-Rad, Herrcules, CA, USA) and examined using an epifluorescence Olympus BX 50 microscope equipped with a 100 W mercury high-pressure bulb and U-MWIB, U-MWIB, and U-MIWG excitation filters. Pictures were taken with an Olympus DP-12 camera (Olympus Optical Co., Hamburg, Germany).

### 2.7. Amplicon-Based Metagenomics and Bioinformatics Analysis (DAS)

This technique analyzed all of the direct samples. When DAS from any direct sample was negative but some other molecular method (qPCR or FISH) had yielded *H. pylori*-positive results, the enriched broth was also tested.

DNA extraction was performed using the GeneJetTM genomic DNA purification kit (Thermo Fisher Scientific, Dreieich, Germany). Directed sequencing was performed on the Illumina MiSeq sequencing platform. Libraries were generated using the primers HS-Forward (5′-CTAATACATGCAAGTCGAACGA-3) [25] and HS2-Reverse (5′-GTGCTTATTCGTTAGATACCGTCAT-3′) [26], which amplify a region of the hypervariable V3-V4 region of the 16S rRNA gene of the *Helicobacteraceae* family, according to the instructions specified in the DNA library preparation guide for sequencing in the Illumina MiSeq sequencer (Amplicon P.C.R., 2013) using the KAPA HiFi HotStart enzyme (Kapabiosystems, Wilmington, MA, USA). Sequencing was performed following the paired-end protocol (2 × 300 bp). The PCR conditions were 95 °C for 3 min, followed by 28 cycles of amplification (denaturation at 95 °C for 30 s, annealing at 55 °C for 30 s, and extension at 72 °C for 30 s), and an additional extension step for 5 min at 72 °C.

Bioinformatic analysis was carried out once the raw sequences were obtained through the free-access platform QIIME 1.9.1 [27]. The corresponding scripts used in the Microbiome Helper virtual box [28] were applied. Forward and reverse sequences were first joined with the PEAR v0.9.19 command [29], and the FastQC tool Version 0.11.9 [30] was used to confirm that the sequences had been joined correctly. Afterward, FASTX-Toolkit v0.014 [31] was applied to filter the stitched reads by length and quality score. A minimum quality value of Q30 over at least 90% of the read or less than 200 bp and reads with any ambiguous base (“N”) were eliminated. Following this protocol, the FastQC tool was used again to remove sequences with low-quality tails.

QIIME wrapper scripts were used to classify the sequences into different taxonomic units (OTUs), defined with a genetic similarity limit of 97%, using the Silva database of high-quality ribosomal RNA Silva_132_release.tgz [32]. Specifically, the selection of OTUs was carried out with the open reference option using the open-source methods SortMeRNA v2.0 [33] and SUMACLUST v1.0.00 [34]. Finally, OTU sequences that were not identified at the species level by using the Silva132 database were aligned against the NCBI database using the online BLAST tool (https://blast.ncbi.nlm.nih.gov/Blast.cgi. Last accessed 31 January 2024).

## 3. Results

### 3.1. H. pylori Culture

After several days of incubation, no presumptive colonies were observed on selective Dent agar from direct sample plating with or without membrane prefiltration. In the enriched samples, some characteristic colonies were covered with a nonspecific bacterial mass, preventing the colonies’ isolation. Isolated colonies and suspicious mixed cultures were collected for identification by qPCR, and all yielded negative results.

### 3.2. H. pylori Detection by qPCR

All the qPCR products from the presumptive positive samples were purified and sequenced. Results indicated that all of them were 99–100% similar to the *H. pylori* VacA gene sequence in GenBank. qPCR analysis detected *H. pylori* in feces from 22 out of the 45 analyzed samples (48.9%) (Figure 3; Table 1), of which 18 were positive both directly and after enrichment. The remaining four samples were positive only after enrichment in Dent culture broth. One sample that yielded positive results for HpSA antigen (H17) was negative by qPCR.

### 3.3. H. pylori Detection by FISH and DVC-FISH

An *H. pylori*-specific HPY probe detected its presence in 19 (42.2%) of the 45 enriched samples. In most of these samples, cells were embedded in a large amount of accompanying microbiota, which made it difficult to detect and identify *H. pylori* by epifluorescence microscopy.

FISH analysis with the same probe after DVC treatment showed elongated cells, viable *H. pylori* cells, in 24 (53.3%) of the 45 direct samples analyzed.

All direct samples were also analyzed with CLAR I-II-III probes to detect viable *H. pylori* cells harboring 23S rDNA mutations that confer resistance to clarithromycin. Positive results were obtained in 17 (37.7%) of the stool samples that had also yielded positive results by DVC-FISH analysis with a LAN-specific probe. No specific DVC-FISH-negative sample was found to be positive by this assay.

All of the samples that yielded positive results for the HpSA antigen test were also positive by FISH. When comparing qPCR and FISH results, PCR was positive in one FISH-negative sample, while in three samples, the FISH technique yielded positive results while PCR was negative (Table 1).

### 3.4. Illumina Deep-Amplicon Sequencing (DAS)

A region of the V3-V4 segment of the 16S rRNA gene of *Helicobacter* spp. was amplified and submitted to deep sequencing through the Illumina MiSeq platform. A total of 2,927,650 raw sequences were obtained. After filtering by quality, joining paired-end reads, and eliminating chimeras, 1,862,312 high-quality sequences remained, grouped into 3170 OTUs (3% cutoff level). The bioinformatic analysis allowed taxonomic assignment down to the genus level (97% homology), and 87 OTU were assigned to *Helicobacter* spp. For identification at the species level, the sequences of each OTU of *Helicobacter* spp. were aligned against the NCBI database by the BLAST online tool, which allowed identifying sequences with 99% and 100% homology with the 16S rRNA of *H. pylori* in 27 samples. In nine samples, sequencing of the direct samples was negative, and *H. pylori* sequences were observed after analysis of the enriched samples.

In addition to these findings, DNA from *H. hepaticus* species, with 99–100% homology to the database sequences, was identified in eight direct samples and three samples after enrichment. In all cases, *H. hepaticus* DNA coexisted with *H. pylori* sequences, except for two samples, H9 and H25, where only *H. hepaticus* sequences were detected directly and after enrichment, respectively (Table 1).

DAS yielded negative results in three positive samples for *H. pylori* by other molecular methods (H11, H14, and H44). On the other hand, sequences belonging to *H. pylori* were detected in 5 samples that were negative by any other molecular technique (H23, H25, H26, H27, H40); two of these (H25, H27) showed the presence of *H. hepaticus* sequences.

## 4. Discussion

Different techniques, such as culture, PCR, or enzyme immunoassays, have been used to detect *Helicobacter pylori* in feces, as the required samples are readily available. These noninvasive tests are becoming vital in clinical practice, especially for children [1]. In this study, the detection and identification of *H. pylori* in the feces of 45 pediatric patients with dyspeptic symptoms were carried out using molecular and culture methods and the HpSA test, previously performed in the hospital (Table 2).

HpSA is a screening method with good sensibility and specificity, especially in proton-pump inhibitor (PPI)-naïve patients. Some authors who have studied the correlation between the urea breath test and the HpSA have reported the usefulness of the HpSA test for the diagnosis of active *H. pylori* infection [35,36].

The culture method is the most specific for detecting viable *H. pylori* cells in clinical samples. Its main advantage is that it allows for the pathogen’s isolation, favoring the appropriate antibiotic treatment choice. However, the success rate of isolating *H. pylori* from stool by culture could be higher. No optimal selective and specific medium has been described for isolating *Helicobacter* spp. from complex samples such as feces [14]. In this work, isolation of the microorganism was not possible due to the overgrowth of the accompanying microbiota, despite the use of Dent agar medium with polymyxin B, which had previously been shown to be quite selective for the detection of *H. pylori* in environmental and clinical samples [37]. It could be due to the conversion of *H. pylori* to its VBNC form in the intestinal tract, where it contacts degradative enzymes, polysaccharides, competent microorganisms, and bile acids [9].

Molecular techniques based on PCR have been previously used for diagnosing *H. pylori* in feces from children [38]. Primers used for this study have previously been shown to be highly specific for *H. pylori* identification [21]. However, the sensitivity of these methods can be low since feces naturally contain PCR inhibitors, which leads to false-negative results [14]. In this work, 18 positive direct samples were obtained by qPCR. After selective enrichment, *H. pylori* DNA was detected in four previously negative samples, suggesting that incorporating a prior enrichment step improves the detection of this slow-growing fastidious pathogen. However, for one sample that was positive by the HpSA test and FISH, qPCR was negative even after enrichment, probably due to the remaining inhibitory compounds in the sample. In one sample that was only positive by qPCR (H22), DAS showed the presence of *H. pylori*, thus confirming the specificity of the qPCR assay.

FISH is another molecular technique that has previously demonstrated its usefulness for detecting *H. pylori* in environmental and clinical samples [17,22]. Through FISH, after enrichment, 19 positive samples were identified. *H. pylori* visualization was complex due to background noise and the overgrowth of accompanying nonspecific microbiota in the sample. It may explain the lower number of positive samples than those obtained by qPCR.

Deep-amplicon-based sequencing (DAS) is a culture-independent method that detects low-abundant bacteria, such as pathogenic species, in complex environments [20]. By targeted sequencing with the Illumina MiSeq platform, *H. pylori* were detected in 26 samples, including 5 that had previously been negative for vacA gene qPCR. DAS also revealed the presence of other important NHPH species in stool samples from the studied pediatric patients. In 10 samples, *H. hepaticus* DNA was detected, demonstrating the coexistence of both species in clinical samples for the first time. *H. pullorum* was also detected in one of the samples analyzed. Our results follow other recent studies on pathogenic species of *Helicobacter* non-*H. pylori* (NHPH) [39]. They demonstrated the presence of some gastric NHPH species in biopsy samples from patients with cancer who were negative for *H. pylori*.

Some studies on the pathogenicity of *H. hepaticus* have supported the hypothesis that this species is a human enterohepatic pathogen associated with liver and biliary tract diseases, and it has been proposed that it may be a risk factor in the progression of liver disease to cirrhosis and hepatocellular carcinoma [11].

*H. pullorum* is a commensal bacterium in the intestine of birds and is present in the feces and biopsies of patients with gastroenteritis, chronic liver disease, and inflammatory bowel disease. This species has been detected in 30% of carcasses and processed chicken meat, and there is consensus on considering it an emerging foodborne pathogen. *H. pullorum* is also associated with Crohn’s disease and cholelithiasis [40].

To our knowledge, only in a study by Tsuchiya et al. [41] was the DAS-based 16S rRNA sequencing methodology used to detect NHPH in bile samples from patients with gallbladder cancer, although all the results were negative. Although NHPH DNA has been detected together with *H. pylori* in biopsies of cancer patients [39], until now, there has been no publication on detecting both species together in human feces.

While humans are almost the only natural host of *H. pylori*, most *Helicobacter* enterohepatic species, such as *H. pullorum* or *H. hepaticus*, have animal hosts. Human infection with these species has been suggested to have a zoonotic origin [42]. Thus, the presence of these species in children’s feces detected by DAS could indicate contamination by contact with animals or even from dietary sources.

Although there are better methods than 16S deep amplicon sequencing analysis to detect specific pathogenetic species, it has demonstrated its utility in clinical microbiology for diagnosing polymicrobial infections in complex samples, which can be challenging to achieve using traditional culture-based methods [43]. In the present work, the DAS technique has shown a good correlation with other molecular techniques. Remarkably, DAS yielded positive results in five samples that were negative when using any other molecular technique, two of which also showed the presence of *H. hepaticus* sequences. More exhaustive studies should be performed to determine whether this result demonstrates that the sensitivity of DAS is superior to that of other methods or whether these results are because the detected sequences are similar to those of other microorganisms that may not be included in the databases.

Despite the promising results obtained by FISH, qPCR, and DAS, these techniques cannot provide information about the viability of the detected *H. pylori* cells. Therefore, combining these methods with specific DVC-FISH could improve the accuracy of the results. Our work revealed the presence of elongated, and thus metabolically active, *H. pylori* cells in 24 out of the 45 analyzed stool samples, which seems to demonstrate that the factors that affect the cultivability of *H. pylori* in feces do not affect its metabolic activity and suggests that the *H. pylori* in these samples are potentially infectious. As reported in the study by Moreno et al. [17], the efficiency of the DVC-FISH technique was higher than that of qPCR, probably due to the composition of samples rich in Taq polymerase inhibitors. The increased size of the cells subjected to the DVC process, together with the lower amount of interfering microbiota in the direct samples, may explain the higher number of positive samples compared to the FISH technique after enrichment.

This technique also allowed for the identification of viable *H. pylori* cells that presented some 23S rRNA mutations that conferred resistance to clarithromycin. We obtained positive results for 37% of the samples. This percentage is relatively superior to that found by a study performed in 18 European countries [44] in 2018, which observed a mean percentage of resistance to clarithromycin of 21% and 17% in Spanish centers. To our knowledge, this is the first time that viable *H. pylori* cells resistant to clarithromycin have been identified in fecal samples from children showing symptoms of infection. This method would provide complementary information before establishing eradication therapy, avoiding the damage caused by the abuse of antibiotics in children and the increase in secondary or multiple resistances.

Finally, it is remarkable that many samples were negative when using the HpSA test but positive when using molecular techniques, suggesting that the antigen test may need to be more vitally accurate. More exhaustive studies, in which more patients are followed over time, should be performed to validate these results.

## 5. Conclusions

The results of this study show the presence of *H. pylori* and other *Helicobacter* spp. in stool samples using molecular techniques, including targeted sequencing and DAS. The DVC-FISH method demonstrated the presence of viable, potentially infective *H. pylori* cells in a high percentage of the children’s stools. We also detected a high number of *H. pylori* strains resistant to clarithromycin. The DVC-FISH technique is, therefore, a noninvasive, relatively rapid (48 h), sensitive, and specific method for detecting viable *H. pylori* cells in feces and may be a technique to use in addition to culture techniques and from biopsy samples for accurate diagnosis.

Finally, to the best of our knowledge, the results obtained by DAS demonstrate for the first time the presence of other NHPH species, such as *H. hepaticus*, in child stool samples, providing a starting point for studying the possible fecal–oral transmission of these species.

## Figures and Tables

**Figure 1 diagnostics-14-01216-f001:**
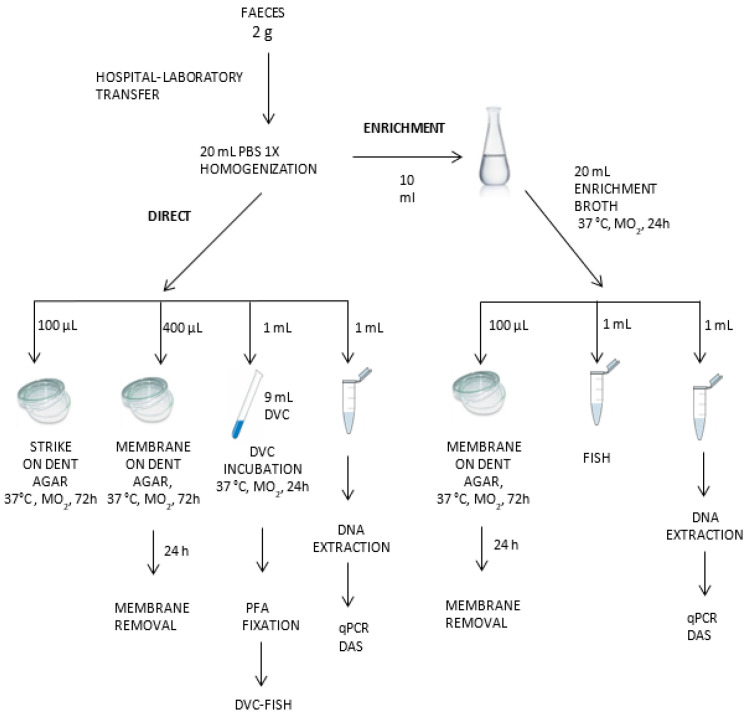
Detection protocol for *H. pylori* in feces by molecular and culture techniques.

**Figure 2 diagnostics-14-01216-f002:**
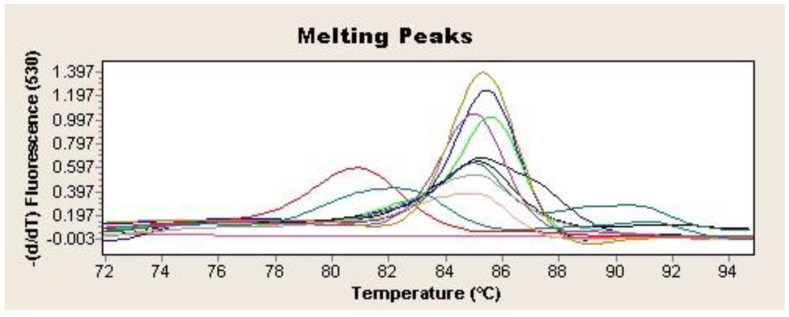
Tm 84.5–86.5 °C for *H. pylori* vacA gene qPCR positive samples. Each sample is represented by a different colour. Negative control is represented in purple (-).

**Figure 3 diagnostics-14-01216-f003:**
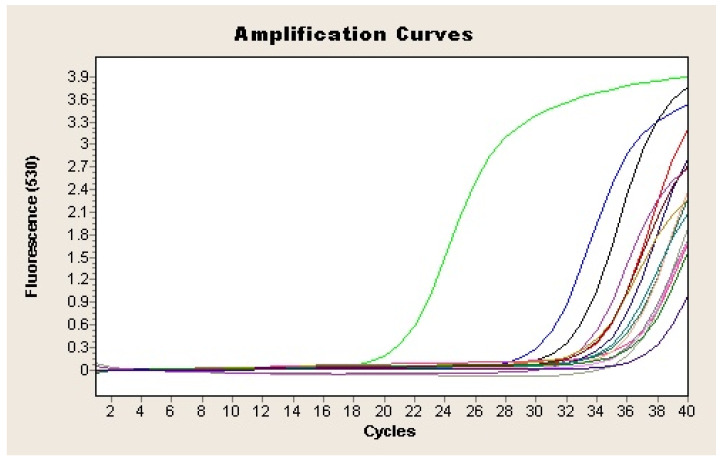
Amplification plot of the vacA gene of *H. pylori* from feces samples. Each sample is represented by a different colour. Negative control is represented in purple (-) and positive sample in green (-).

**Table 1 diagnostics-14-01216-t001:** Detection of *H. pylori* by qPCR, FISH, DVC-FISH, and DAS in child feces samples.

Sample	HpSA Test	Treatment Prior to Analysis	qPCR	*H. pylori* FISH	*H. pylori* DVC-FISH	*H. pylori* DVC-FISH (CLAR)	DAS
**H1**	−	Direct	+		+	+	*H. pylori*, *H. hepaticus*
		Enriched	+	+			ND
**H2**	+	Direct	+		+	+	*H. pylori*, *H. hepaticus*
		Enriched	+	+			ND
**H3**	−	Direct	−		+	+	-
		Enriched	+	+			*H. pylori*, *H. hepaticus*
**H4**	−	Direct	+		+	+	-
		Enriched	+	+			*H. pylori*, *H. hepaticus*
**H5**	+	Direct	+		+	+	*H. pylori*, *H. hepaticus*
		Enriched	+	+			ND
**H6**	−	Direct	+		+	+	*H. pylori*, *H. hepaticus*
		Enriched	+	+			ND
**H7**	−	Direct	+		+	+	*H. pylori*
		Enriched	+	+			ND
**H8**	−	Direct	−		+	−	*H. pylori*
		Enriched	+	−			ND
**H9**	−	Direct	+		+	+	*H. hepaticus*
		Enriched	+	−			ND
**H10**	−	Direct	+		+	+	-
		Enriched	+	−			*H. pylori*, *H. pullorum*
**H11**	−	Direct	+		+	+	-
		Enriched	+	−			ND
**H12**	−	Direct	−		−	−	-
		Enriched	−	−			ND
**H13**	−	Direct	−		+	+	*H. pylori*
		Enriched	−	+			ND
**H14**	−	Direct	−		+	−	-
		Enriched	−	+			ND
**H15**	+	Direct	−		+	+	*H. pylori*
		Enriched	+	−			ND
**H16**	−	Direct	−		−	−	-
		Enriched	−	−			ND
**H17**	+	Direct	−		+	+	*H. pylori*
		Enriched	−	+			ND
**H18**	−	Direct	−		−	−	-
		Enriched	−	−			ND
**H19**	−	Direct	−		−	−	-
		Enriched	−	−			ND
**H20**	−	Direct	−		−	−	-
		Enriched	−	−			ND
**H21**	−	Direct	−		−	−	-
		Enriched	−	−			ND
**H22**	−	Direct	+		−	−	*H. pylori*
		Enriched	+	−			ND
**H23**	−	Direct	−		−	−	*H. pylori*, *H. hepaticus*
		Enriched	−	−			ND
**H24**	−	Direct	−		−	−	-
		Enriched	−	−			ND
**H25**	−	Direct	−		−	−	-
		Enriched	−	−			*H. hepaticus*
**H26**	−	Direct	−		−	−	ND
		Enriched	−	−			*H. pylori*
**H27**	−	Direct	−		−	−	*H. pylori*, *H. hepaticus*
		Enriched	−	−			ND
**H28**	−	Direct	−		−	−	-
		Enriched	−	−			ND
**H29**	−	Direct	−		−	−	-
		Enriched	−	−			ND
**H30**	−	Direct	−		+	−	*H. pylori*
		Enriched	+	+			ND
**H31**	−	Direct	−		−	−	-
		Enriched	−	−			ND
**H32**	+	Direct	+		+	+	*H. pylori*, *H. hepaticus*
		Enriched	+	+			ND
**H33**	−	Direct	+		+	−	*H. pylori*
		Enriched	+	+			ND
**H34**	+	Direct	+		+	+	-
		Enriched	+	+			*H. pylori*
**H35**	−	Direct	−		−	−	-
		Enriched	−	−			ND
**H36**	+	Direct	+		+	+	*H. pylori*
		Enriched	+	+			ND
**H37**	−	Direct	−		−	−	-
		Enriched	−	−			ND
**H38**	+	Direct	+		+	+	*H. pylori*
		Enriched	+	+			ND
**H39**	−	Direct	−		−	−	-
		Enriched	−	−			*H. pylori*
**H40**	−	Direct	−		−	−	*H. pylori*
		Enriched	−	−			ND
**H41**	−	Direct	+		+	−	-
		Enriched	+	+			*H. pylori*
**H42**	−	Direct	+		+	−	-
		Enriched	+	+			*H. pylori*
**H43**	−	Direct	−		−	−	-
		Enriched	−	−			ND
**H44**	−	Direct	+		+	−	-
		Enriched	+	+			ND
**H45**	−	Direct	−		−	−	-
		Enriched	−	−			*H. pylori*

ND: Not determined.

**Table 2 diagnostics-14-01216-t002:** Results of *H. pylori* detection in child feces samples according to the different techniques used in this work.

Method	HpSA Test	Direct qPCR	qPCR after Enrichment	*H. pylori* FISH	*H. pylori* DVC-FISH	*H. pylori* DVC-FISH (CLAR)	DAS
Number of positive samples/Total samples (%)	8/45 (17.8%)	18/45 (40%)	22/45 (48.9%)	19/45 (42.2%)	24/45 (53.33%)	17/45 (37.8%)	27/45 (60%)

## Data Availability

Data are contained within the article.

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
