# Peer review of "Use of Deep-Amplicon Sequencing (DAS), Real-Time PCR and In Situ Hybridization to Detect H. pylori and Other Pathogenic Helicobacter Species in Feces from Children"

_diagnostics, 2024, doi:10.3390/diagnostics14121216_

Round 1
Reviewer 1 Report
Comments and Suggestions for Authors
In this manuscript, the authors aimed to present a non-invasive way to quantify Helicobacter species in human feces, particularly child feces. The last author of this paper seems to be a specialist of H. pylori detection (6 papers mentioned in the 44 in bibliography). But, I found only one paper of the corresponding author regarding this subject (PMID: 32393934). I am quite surprised.
The study is, to my point of view, correct, but I have some comments.
1/ To illustrate purpose of the author in the introduction, reference 8 is fine but a recent meta-analysis is probably better. I let the authors judge (PMID: 36645421). Regarding this analysis, fecal-oral transmission is the most common route for H. pylori.
2/ Is the fecal treatment method already published?
3/ For the FISH method you used a correct positive control. But, why a negative control has not been also used? It is possible to have an off-target with FISH method.
4/ For the real time Q-PCR you used 40 cycles which is correct. But which threshold for positive and negative results did you used? Indeed, depending of this threshold you will find less than 22 positive samples. To my point of view, you have less than 22 positive samples with your Q-PCR.
5/ In table 2, there is problem of percentages. Calculations are not good.
6/ I prefer the conclusion of your manuscript than the conclusion of your abstract: “These data provide evidence of fecal-oral transmission of H. pylori and suggest possible fecal-oral transmission of other pathogenic Helicobacter species.”. You have a possible evidence of fecal-oral transmission of H.pylori and only a possible evidence. Not something sure.
Your manuscript is correct but some points have to corrected. Mainly the absence of negative control, the threshold and the calculations in table 2.
Author Response
In this manuscript, the authors aimed to present a non-invasive way to quantify Helicobacter species in human feces, particularly child feces. The last author of this paper seems to be a specialist of H. pylori detection (6 papers mentioned in the 44 in bibliography). But, I found only one paper of the corresponding author regarding this subject (PMID: 32393934). I am quite surprised.
Dr. Miguel Tortajada Girbés is considered a reference in the field of Pediatrics and its specific areas. Through his high level of knowledge and practical skills, he has held positions within the specialized services of prestigious health institutions. Since May 2022, and after a public competition, he has been appointed current Head of Section of the Pediatric Pneumology and Allergy Unit of the Hospital Universitario y Politécnico La Fe de Valencia, the most important hospital of the city. He is a member of the European Academy of Allergology and Clinical Immunology and the Spanish Association of Pediatrics. As a result of his interest in all areas of paediatrics, Dr. Tortajada was the promoter of this collaborative work on the diagnosis of H. pylori infection in children and was responsible for collecting all the clinical data. His help in carrying out this work, as well as the clinical perspective he provided during its development, justifies his participation as “corresponding author”.
The study is, to my point of view, correct, but I have some comments.
1/ To illustrate purpose of the author in the introduction, reference 8 is fine but a recent meta-analysis is probably better. I let the authors judge (PMID: 36645421). Regarding this analysis, fecal-oral transmission is the most common route for H. pylori.
We agree that a meta-analysis is better than the reference we provided. Now we have changed it by the one kindly suggested by the referee.
2/ Is the fecal treatment method already published?
We have previously published this protocol for detecting H. pylori from environmental and feces samples (i.e. doi: 10.1007/s10123-020-00135-z; https://doi.org/10.1016/j.envpol.2020.114768;)
3/ For the FISH method you used a correct positive control. But, why a negative control has not been also used? It is possible to have an off-target with FISH method.
We are sorry, we unintentionally omitted this point in the manuscript. As a negative control, in all the assays we used a feces sample that had yielded previous negative results (PCR for VacA and 16SrRNA gene; FISH with specific H. pylori and Helicobacter sp. probes and culture) for Helicobacter presence. Now we have added it to the manuscript.
4/ For the real time Q-PCR you used 40 cycles which is correct. But which threshold for positive and negative results did you used? Indeed, depending of this threshold you will find less than 22 positive samples. To my point of view, you have less than 22 positive samples with your Q-PCR.
Samples were considered positive for H. pylori if the cycle threshold (Ct) was less than or equal to 40 and showed Tm values between 84.8 and 86.5. To confirm the identification of the amplicons, the PCR product from the 22 presumptive positive samples were purified and sequenced. Results indicated that all of them were 99–100% similar to H. pylori VacA gene sequence in GenBank. Now, this information has been clarified in our manuscript (“Material and Methods” and “Results”).
5/ In table 2, there is problem of percentages. Calculations are not good.
According to referee’s comment, we have changed three of the percentages that were wrong. Calculations are good now.
6/ I prefer the conclusion of your manuscript than the conclusion of your abstract: “These data provide evidence of fecal-oral transmission of H. pylori and suggest possible fecal-oral transmission of other pathogenic Helicobacter species.”. You have a possible evidence of fecal-oral transmission of H. pylori and only a possible evidence. Not something sure.
According to referee’s comment and suggestion, we have changed the conclusion of the abstract.
Your manuscript is correct but some points have to corrected. Mainly the absence of negative control, the threshold and the calculations in table 2.
Reviewer 2 Report
Comments and Suggestions for Authors
This article is well-written and easy to understand.
The methodology is reproducible.
It is a descriptive research that compares different methodologies for the detection of H. pylori. Although there are many methodologies, which adds great value to this study, the small number of samples does not allow for complex analyses. In this regard, the conclusion that "the DVC-FISH technique may be an alternative to biopsy culture techniques" lacks sufficient evidence; indeed, it could be a technique to use in addition to biopsy culture. Proposing it as an alternative requires additional studies to determine the accuracy of the results of both tests.
Besides this, some aspects of the form need to be reviewed. I cite the following:
Line 177: A point is missing after "Table 1"
Line 233: The title of that figure is not appropriate; what is listed as the title should be part of the figure explanation.
Line 322: "H. pylori" should be italicized. This figure requires a brief explanation.
Lines 350, 351, 402: "H. pylori" should be italicized.
Author Response
This article is well-written and easy to understand.
The methodology is reproducible.
It is a descriptive research that compares different methodologies for the detection of H. pylori. Although there are many methodologies, which adds great value to this study, the small number of samples does not allow for complex analyses. In this regard, the conclusion that "the DVC-FISH technique may be an alternative to biopsy culture techniques" lacks sufficient evidence; indeed, it could be a technique to use in addition to biopsy culture. Proposing it as an alternative requires additional studies to determine the accuracy of the results of both tests.
We totally agree with referee’s comment and now the text has been changed as kindly suggested in the revision.
Besides this, some aspects of the form need to be reviewed. I cite the following:
Line 177: A point is missing after "Table 1"
Line 233: The title of that figure is not appropriate; what is listed as the title should be part of the figure explanation.
Line 322: "H. pylori" should be italicized. This figure requires a brief explanation.
Lines 350, 351, 402: "H. pylori" should be italicized.
The aspects of the form kindly suggested by the referee have now been reviewed. Text has been changed. Moreover, all the missing points after “Figures” and “Tables” have been corrected. Also, all the “H. pylori” have been italicized.
Round 2
Reviewer 1 Report
Comments and Suggestions for Authors
Dear authors,
To my point of view you did the changes I asked to your manuscript.
I thank you.
Best,
Author Response
AUTHORS´ RESPONSE TO THE EDITOR´S COMMENTS
The decision is Accept after minor revision but:
1/The huge sections INTRODUCTION and DISCUSSION are striking, which need to be reduced by 2-3 times.
According to editor recommendations, The Discussion and Introduction sections have been considerably reduced.
2/The huge table 1 in the MATERIALS AND METHODS section causes
amazement and irritation. The table should either be removed or try to
display the data of 1-2 patients in graphical mode.
According to editor recommendations, Table 1 has been finally removed.
3/In the DISCUSSION section I would like to see a clearer understanding of the feasibility of defining Non-pylori Helicobacters.
The following text has been added to the discussion section (lines 353-359):
“Although H. pylori is the best known and characterized pathogenic species, other species also possess potential to cause disease in humans and are collectively referred to as “Non-H. pylori Helicobacters (NHPH). Moreover, their search is usually not routinely performed. In most cases, the histological diagnosis can be confused with that of H. pylori infection, as the only differential feature is the little morphological difference that may exist between the different species [10]”.